# Digenic Inheritance in Rare Disorders and Mitochondrial Disease—Crossing the Frontier to a More Comprehensive Understanding of Etiology

**DOI:** 10.3390/ijms25094602

**Published:** 2024-04-23

**Authors:** Christiane M. Neuhofer, Holger Prokisch

**Affiliations:** 1Institute of Human Genetics, University Medical Center, Technical University of Munich, Trogerstr. 32, 81675 Munich, Germany; 2Institute of Neurogenomics, Computational Health Center, Helmholtz Centre Munich Neuherberg, Ingolstädter Landstraße 1, 85764 Oberschleißheim, Germany; 3Institute of Human Genetics, Salzburger Landeskliniken, University Hospital of the Paracelsus Medical University, Müllner Hauptstraße 48, 5020 Salzburg, Austria

**Keywords:** digenic inheritance, mitochondrial disorders, molecular genetics

## Abstract

Our understanding of rare disease genetics has been shaped by a monogenic disease model. While the traditional monogenic disease model has been successful in identifying numerous disease-associated genes and significantly enlarged our knowledge in the field of human genetics, it has limitations in explaining phenomena like phenotypic variability and reduced penetrance. Widening the perspective beyond Mendelian inheritance has the potential to enable a better understanding of disease complexity in rare disorders. Digenic inheritance is the simplest instance of a non-Mendelian disorder, characterized by the functional interplay of variants in two disease-contributing genes. Known digenic disease causes show a range of pathomechanisms underlying digenic interplay, including direct and indirect gene product interactions as well as epigenetic modifications. This review aims to systematically explore the background of digenic inheritance in rare disorders, the approaches and challenges when investigating digenic inheritance, and the current evidence for digenic inheritance in mitochondrial disorders.

## 1. Introduction

Molecular genetic diagnostics of rare hereditary disorders are conventionally conducted under a monogenic disease model, assuming disease-causing variants to be rare, of high functional impact, and in one disease-determining gene. Due to the large effect size expected from such a variant, sufficient statistical power to identify the disease cause can be achieved in relatively small cohorts. Considering the success of this viewpoint, which enabled the discovery of numerous novel disease-associated genes in the past decades, it is not surprising that monogenic inheritance still shapes our view of the genetics of rare diseases [1].

The monogenic model alone, however, has shown limitations in explaining phenomena such as phenotypic variability, reduced penetrance, and variable expressivity [2]. Furthermore, using the current gold standard methods of Whole-Exome-Sequencing (WES) or Whole-Genome-Sequencing (WGS), diagnostic rates cap at 30–50%, leaving >50% of patients with a suspected hereditary disorder without a diagnosis [3,4,5]. It is unlikely that this diagnostic gap can be bridged by improving variant detection and variant interpretation alone. Increasing knowledge and international collaboration is leading to the creation of larger, well-characterized patient cohorts that allow us to broaden our perspective and look beyond the scope of Mendelian inheritance to identify more complex genetic patterns that underlie or contribute to rare inherited diseases. Genetic patterns determining a phenotype may be more intricate and may involve the interplay of variants in multiple genes.

The simplest form of a complex inheritance pattern is digenic inheritance, referring to an interaction of variants in two genes resulting in a phenotype or disease. This may, depending on the definition, refer to classic digenic (also referred to as true digenic) inheritance, i.e., a disease caused by variants in two genes which cannot individually produce a phenotype, or a monogenic disorder with a modulating variant significantly altering the expected phenotype [6]. While these definitions appear to be clearly distinct, the border between both is often blurry in practice. Both instances can, however, clearly be distinguished from the co-occurrence of multiple diagnoses, resulting in a phenotype comprising symptoms of both or several underlying independent diseases [7]. While digenic inheritance leads to a phenotype that cannot be explained by the individual gene defects but is based on their functional interaction, in cases of double or multiple diagnoses, there is no functional link between the genes involved and the presenting phenotype is a composite addition of several disorders (see Figure 1).

## 2. Pathomechanisms of Digenic Inheritance in Rare Disorders

Digenic causes of disease are the result of an interaction between variants in two genes, adding up to cause or modify a disease or phenotype. These additive effects can result from direct or indirect interaction of gene products or gene-modifying effects of a gene product.

### 2.1. Direct Interaction of Gene Products

Direct interaction between gene products can occur through involvement in the same protein complex. In this case, each variant impacts complex activity individually, but only when combined do they then reach a disease-causing threshold (see Figure 2a).

This is exemplified by the retinopathia pigmentosa (RP)-associated genes *PRPH2* (old denomination: *RDS*) and *ROM1*. *PRPH2* is known to cause monogenic autosomal-recessive (AR) and autosomal dominant (AD) RP and macular dystrophy. It forms homo-oligomers as well as hetero-oligomers with *ROM1.* The complexes are essential for photoreceptor outer segment morphogenesis [8,9]. While *ROM1* has not been associated with monogenic RP, double heterozygosity for variants in *PRPH2* and *ROM1* has been reported to cause digenic RP. *PRPH2*/*ROM1*-associated digenic RP is the earliest example of a digenic disorder, with the first description dating back to 1994 [10]. Another example of a direct interaction of gene products causing digenic interplay resulting in a phenotype is *CDH23*/*PCDH15*-associated Usher syndrome, an illness manifesting with the hallmark symptoms of congenital hearing loss and progressive retinitis pigmentosa. *CDH23* and *PCDH15* code for calcium-adhering transmembrane glycoproteins, that have been shown to directly interact in sensory hearing cells in order to form tip links, which are essential for mechanoelectrical transduction in hearing [11]. Both genes are linked to AR Usher syndrome, but double heterozygosity has also been shown to cause a digenic inherited form [12].

### 2.2. Indirect Link of Gene Product Function

The interaction of gene variants can also be the result of an indirect link of gene product function, for example, convergence on the same or a complementary function or within the same pathway. This is exemplified by the digenic Bartter syndrome affecting the two CLCNKA and CLCNKB chloride channels. Bartter syndrome is a metabolic disorder caused by an impairment of salt reabsorption in the ascending limb of the loop of Henle, resulting in renal salt wasting. Most types of Bartter syndrome follow an AR mode of inheritance, including cases due to biallelic loss-of-function (LoF) variants in *CLKNB* (Bartter syndrome type 3). Bartter syndrome type 4 is a disease in which both CLCNKA and CLCNKB are affected and was found to be associated with biallelic pathogenic mutations of the *BSDN* gene, coding for the essential Barttin subunit of both the CLCNKA and CLCNKB chloride channels [13,14]. As a result, in addition to renal salt wasting, sensorineural hearing loss is a hallmark sign of Bartter syndrome type 4. Notably, a digenic inherited Bartter syndrome type 4 has been reported in patients carrying biallelic predicted loss-of-function (pLoF) mutations in both the *CLKNA* and *CLKNB* genes, respectively [15,16] (see Figure 2b).

### 2.3. Epigenetic Modification—FSHD

While the previously mentioned cases of digenic disorders are caused by direct or indirect interactions of gene products, digenic facioscapulohumeral dystrophy (FSHD) presents an example of an epigenetic modification in a complex disease model, widening the perspective on possible underlying mechanisms. 

FSHD is a muscular disorder characterized by a weakness of the facial, scapular, and upper arm musculature. Its monogenic inheritance pattern is autosomal dominant, and it is associated with the reduction in copy number and loss of epigenetic silencing of the D4Z4 repeat array located in 4q35, resulting in unphysiological expression of the transcription factor double homeobox 4 (DUX4). The expression of DUX4 is highly regulated and mainly restricted to the early steps of embryonic development. The expression is suppressed by hypermethylation. There are two major 4q alleles (4qA and 4qB), but only the 4qA allele is expressed in skeletal muscle. Pathologic expression of DUX4 only occurs if both a hypomethylation of the D4Z4 locus and a permissive haplotype 4qA coincide. Digenic inheritance is observed when hypomethylation is caused by pathogenic mutations in epigenetic modifier genes, such as *SMCHD1* [17], *DNMT3B* [18], and most recently, in *LRIF1* [19] (see Figure 2c). FSHD was the first neurological disorder with a well-established digenic background and about 5% of the FSHD cases are digenic. Digenic impact by epigenetic modifiers may be harder to detect in extremely rare disorders where gene expression modification is less well characterized. 

## 3. Investigating Digenic Inheritance—Approaches and Challenges

There is increasing evidence of digenic inheritance in rare diseases, but we are only at the beginning and the full potential is far from being exhausted. In the following, we will look into the approaches and methods for studying the contribution of two or more genes to the development of a rare disorder as well as the challenges and limitations to consider. To put these methods into perspective, we will have a look at variant evaluation in current monogenic, rare disorders and multifactorial, frequent disorders (Table 1). 

### 3.1. Variant Evaluation in Monogenic Disorders

Interpretation of molecular genetic results under a monogenic disease model involves analyzing various aspects of a genetic variant to understand its potential significance in a structured and standardized way. The American College of Medical Genetics and Genomics (ACMG) guidelines provide a framework for classifying variants as pathogenic, likely pathogenic, uncertain significance, likely benign, or benign based on several lines of evidence [20]. Aspects the ACMG guidelines consider for making this distinction include the following:

#### 3.1.1. Population Data

Variants found at high frequencies in population databases (e.g., gnomAD, 1000 Genomes Project) are less likely to be causative of a rare disorder. Under a monogenic model, disease-causing variants are expected to be rare, in line with the disease incidence. However, variant frequency can differ in different population groups, for example, due to heterozygous advantage or the presence of founder mutations.

#### 3.1.2. Inheritance Patterns

Segregation analysis examines if observed variants fit the inheritance pattern expected based on family history or the suspected disorder. For example, segregation with disease within a family or the de novo occurrence of a variant with an expected dominant inheritance mechanism supporting pathogenicity. 

#### 3.1.3. Functional Data

Experimental studies that directly demonstrate the impact of a variant on gene function can strongly support pathogenicity. Variant databases, such as ClinVar [21], provide curated variant information from various sources, including clinical laboratories, research studies, and expert reviews. In the absence of variant-level functional information, understanding the gene function and an association with known diseases or biological pathways relevant to the observed phenotype can improve interpretation. Variants occurring in known and essential functional domains of a gene are more likely to be pathogenic.

#### 3.1.4. Curated Knowledge Databases

Interpretation of monogenic variants profits from a great number of previous studies on gene–phenotype associations. This body of knowledge is accessible in curated clinical databases, such as OMIM [22] and GeneReviews [23], and can be leveraged to help interpret the clinical relevance of a variant.

#### 3.1.5. Computational Prediction Tools

In silico prediction refers to the use of computational tools to estimate the impact of a variant on function based on factors such as conservation, protein structure, and sequence features to assess the likelihood of pathogenicity. Examples include SIFT, PolyPhen-2, CADD, and REVEL. Pathogenicity is considered more likely if multiple tools consistently predict deleterious effects.

### 3.2. Statistical Approach in Frequent, Multifactorial Disorders

Many common disorders, such as coronary heart disease and high blood pressure, are known to be multifactorial disorders. A multifactorial disease cause, in the classical sense, encompasses the genetic background which is determined by the presence of multiple variants with small, additive detrimental or protective effects, resulting in an underlying disease risk and additional non-genetic factors that have an impact on disease manifestation and presentation. These factors can be diverse and numerous, including lifestyle factors, like nutrition and activity, and exposure, such as working conditions or environmental features. The more complex a disease model, the harder it is to discern the impact of individual contributing factors. Multifactorial disorders therefore rely on statistical modeling of additive factors in large patient cohorts, rather than on the evaluation of individual variants in a single patient, as in monogenic disorders. 

In common disorders, complex genetic traits are typically explored in genome-wide association studies (GWAS) using genotyping arrays to detect genetic risk or protective factors [24]. Genotyping arrays have been designed to detect single nucleotide polymorphisms (SNPs) with allele frequencies of >1 or >5% to gain statistical power in a GWAS study for individually relatively small effect sizes [25]. This limits the potential to detect lower-frequency variants with an intermediate impact in an oligogenic disease model [26] and restricts applicability of GWAS to complex inheritance in rare disorders, such as mitochondrial disease.
ijms-25-04602-t001_Table 1Table 1Approaches in genetic analysis based on the assumed inheritance.
MonogenicDigenicMultifactorialPopulation Datarare variants, MAF in keeping with disease incidencecombined MAF of variants in 2 genes in keeping with disease incidence expected to be attributable to digenic causecommon variants(MAF > 1%, >5%)Inheritance Patternsegregation in pedigree with the diseasesegregation analysis is applicablenot applicable due to frequency of the investigated variants and smaller effect sizesFunctional Datafunctional studies as the gold standardfunctional studies challenging to conduct, but plausible mechanism of digenic interaction necessaryusually absentCurated Databasese.g., OMIM [15], GeneReviews [16]e.g., OMIM [15], OLIDA [25]e.g., GWAS CatalogComputational Predictioncomputational prediction based on knowledge (e.g., protein structure, sequence features, amino acid conservation)novel, machine learning powered prediction toolsestimated effect size attributable to a locusStatistical Analysisburden testingadaptive burden testing, novel statistical toolsGWAS

### 3.3. Variant Evaluation in Digenic Disorders

Current studies often lack the power to detect digenic variant effects in a genome-wide screen. Therefore, the suspicion of a digenic disease cause often starts from single or few observations that stand out in effect size in stringently characterized cohorts and well-investigated phenotypes. In practice, this may be, for example, the detection of variants in two genes known to be associated with a certain phenotype, or acting in the same protein complex, clinically fitting the patient’s disease or an unexpected phenotype in a presumed monogenic disease with a mostly homogeneous presentation. Whichever may be the observation leading to the consideration of a digenic inherited disorder, the next step must be a validation of the finding. 

Collecting genetic, statistical, and functional underpinnings when studying hereditary disease is not a novel approach but has, in a similar way, formed our knowledge of genetics up until now. While the lines of evidence stay, in essence, the same, the way to look at data must be adapted when considering digenic inheritance.

#### 3.3.1. Population Data 

Frequency data for variants in established population databases such as gnomAD can be referred to for evaluating variants when suspecting digenic disease. The probability of co-occurrence of deleterious variants in two digenic candidate genes can be calculated considering the combined minor allele frequencies (MAF) of deleterious variants in the respective genes and should be in line with the expected frequency of the respective disease or phenotype in question.

#### 3.3.2. Inheritance Patterns

On the pedigree level, statistical evidence can be provided by segregation analysis, and gains in strength in large pedigrees with multiple affected persons. In digenic disease, in-depth phenotyping is important to interpret the observed patterns correctly. This is especially true for the question of genetic modifiers to explain phenotypic variability. 

The expected pattern depends on the suspected digenic mechanism (modifying or synergistic effect). For example, a maternally inherited mitochondrial variant associated with a maternal phenotype may be modified by a paternally inherited variant in a nuclear gene, leading to an altered phenotype in a child. In the case of suspected classic digenic inheritance, in contrast, both parents would be expected to be asymptomatic (Figure 1).

#### 3.3.3. Statistical Approaches

Given a large enough cohort, the question of digenic inheritance contributing to a specific phenotype can also be approached with statistical methods. Gene burden testing can determine the contribution of rare variants in a given set of genes to a rare disorder. This method has, however, been shown to exhibit limited power in diseases with strong locus heterogeneity, and the presence of both protective and detrimental variants. To overcome these limits, methodically modified ‘adaptive burden tests’ have been developed that are more robust to these factors [27,28]. 

There is an effort to develop powerful statistical tools, specific to the digenic question, for application in large cohorts and datasets [29,30]. With WES and WGS becoming the gold standard methods for the diagnostics of hereditary, rare disorders, digenic inheritance studies may evolve from small scale studies, relying on single pedigrees or small case series towards larger scale analysis, leveraging the growing body of knowledge and available tools.

#### 3.3.4. Functional Data 

In digenic disorders, a suitable functional model must reproduce a phenotype with genetic defects in two distinct genes and the absence of this phenotype in presence of just one of both. The complexity of the interactions of these synergistic effects to produce a molecular defect renders functional analysis a challenge and gets more complicated even in oligo- or polygenic settings. Functional evidence is therefore, in practice, often limited or omitted in reports of digenic inheritance. Nonetheless, a plausible mechanism of interaction needs to be established to support a hypothesis of digenic interaction. Genetic findings should be correlated with the clinical phenotype to understand how the suspected gene–gene interactions can contribute to the manifestation of the rare disorder. 

In turn, reproducible detection of digenic interplay between genes may also inform on function and alert to a possible not yet characterized functional link. 

#### 3.3.5. Curated Knowledge Databases

With the accumulation of disseminated reports of digenic inheritance, often in case reports with one or few pedigrees, the need to document this growing body of knowledge in a publicly available database has arisen. The OMIM database [22] documents mode of inheritance and denominates gene combinations in diseases with reported digenic inheritance as DR/DD (digenic recessive and digenic dominant). DIDA (digenic diseases database) was launched in 2015 in an effort to systematically collect and curate published knowledge on digenic inherited traits [31]. This endeavor is impacted by the limited evidence reports of digenic inheritance often provided. The database has been overhauled to accommodate higher degree oligogenicity and renamed to OLIDA [32]. ORVAL (oligogenic resource for variant analysis), an additional oligogenicity centered platform integrating various prediction tools, has been launched to allow for the detection of candidate variant combinations based on interaction [33]. 

#### 3.3.6. Computational Prediction Tools

Machine learning (ML), due to its power to detect complex patterns and relationships in big datasets, has become a popular approach to develop specific prediction algorithms to address complex inheritance [34]. Key advantages of applying ML to the question of digenic inheritance include the possibility of integrating different types of data (sequencing, functional, clinical), leveraging the full potential of available large datasets, and accounting for genetic and phenotypic heterogeneity. Using the DIDA dataset, for example, VarCoPP [35], DiGePred [36], OligoPVP [37], and the Digenic Effect Predictor [6,38] were developed. These prediction tools will likely adapt and improve over time with the growing body of knowledge to learn from and to train ML algorithms on. 

With the large amount of resources to approach the question of digenic disorders appearing in high frequency over the last years, it is apparent that the challenge lies at the interface of bioinformatics, medicine, and biology, namely the application of novel bioinformatic methods with genetic data from well-phenotyped, undiagnosed cases and the validation of resulting candidates. Currently, there is no commonly consented framework for variant interpretation under a digenic inheritance model, like the ACMG criteria in monogenic disorders. Papadimitriou et al. have, after curation of the OLIDA database for oligogenic inheritance, outlined several recommendations for reporting variants in an oligogenic disease model based on their experience with existing reports [35].

## 4. Digenic Inheritance in Mitochondrial Disorders

Several characteristics make mitochondriopathies particularly interesting for investigating digenic disease. Mitochondrial function has been studied and characterized in depth on a molecular level. Genetically, mitochondrial disease shows vast locus heterogeneity with more than 400 currently known disease genes [39]. About 1500 gene products interact to fulfill the various mitochondrial functions in mostly well-characterized complexes and pathways—45 known disease genes alone contribute to complex I function as a subunit or assembly factor for example. As a functional link is a prerequisite to digenic additive effects of variants, we propose to take advantage of the known interactions and screen for co-occurring variants in interacting genes as a functional entity. 

### 4.1. Phenotypic Spectrum as an Indicator of Genetic Modifying Factors

In addition to genetic heterogeneity, mitochondrial disorders show a broad range of phenotypes, even with a single underlying mutation. A possible and often cited explanation for this is the degree of heteroplasmy, i.e., the fraction of mitochondrial DNA (mtDNA) carrying a certain variant, which can differ greatly between individuals as well as between tissues from the same individual. However, degree of heteroplasmy alone falls short to explain the phenotypic diversity we observe. This has been shown in mtATP6-associated disorders [40,41] as well as in m.3243A>G (MT-*TL*)-associated phenotypes [42]. 

The variant m.3243A>G is a relatively common cause of mitochondrial disorder with a population frequency estimated around 0.02–0.2% in clinically unselected cohorts [43,44]. While it was first described as the cause of mitochondrial myopathy, encephalopathy, lactic acidosis, and stroke-like episodes (MELAS) [45,46], patients have been presenting more frequently with diabetes mellitus and hearing impairment (maternally inherited diabetes mellitus and deafness, MIDD) [47,48]. Since the initial description, a broad range of phenotypic presentations has been described, ranging from asymptomatic carriers to severe multisystem disorder that can encompass endocrine, sensory, renal, neurologic, cardiac, and intestinal symptoms of variable severity.

From a clinical perspective, this pronounced phenotypical variability prompts us to move from the view of a fixed disease entity to the view of a disease spectrum. Furthermore, from a pathomechanistic perspective, this observation encourages us to search modifying factors for a possible underlying cause. Digenic or even oligogenic mechanisms could be a possible explanation. Indeed, phenotypic studies in m.3243A>G carriers showing evidence of high heritability for specific traits support the possible modifying effect of nuclear genetic factors [49].

In keeping with this hypothesis, a linkage study with 488 probands carrying the same m.3243A>G variant, and presenting clinically homogeneous phenotypes segregating in distinct families, found evidence for nuclear factors impacting the associated phenotype (see Figure 3a) [50]. In the identified chromosomal regions, Boggan et al. highlighted *MTERF1*, *PMPBC*, *ATP1A3* and *ADCK3* as candidate genes that may act as phenotypic modifiers and were able to identify phenotypic traits, such as encephalopathy and stroke-like episodes, that seem to be more strongly determined by a nuclear genetic background compared to others in m.3243A>G carriers [50]. 

### 4.2. Digenic Inheritance in Mitochondrial Disorders with Reduced Penetrance

Reduced penetrance can be viewed as an extreme of phenotypic variability and the asymptomatic end of a disease spectrum. Impacting factors can be suspected when comparing the stark contrast of an asymptomatic carrier to a patient exhibiting a clear phenotype. 

An example is the reversible infantile respiratory chain deficiency (RIRCD), a severe, potentially fatal, infantile-onset disease. A remarkable characteristic of this disease is the potential of spontaneous full recovery for children who survive the initial severe phase [51]. RIRCD is associated with the homoplasmic m.14674T>C variant in MT-*TE*, coding for mitochondrial Glutamate tRNA. While the disease is extremely rare, with less than 100 cases described worldwide, the variant m.14674T>C is present in the general population, with about 0.005% being carriers (gnomAD: 0.004% (homoplasmic), HelixMTdb: 0.006% (homoplasmic)) [52]. Therefore, a digenic disease model has been proposed in RIRCD, which endeavors to explain the observed severely reduced penetrance. 

In their study, Hathazi et al. found rare variants in genes linked to mt-tRNAGlu and mt-tRNAGln modification or Glutamate/Glutamine metabolism (*EARS2*, *TRMU*, *QRSL1*, *GOT2*, *GLS*, *MSS51*) in 24/27 investigated affected individuals but not in 15 analyzed healthy family members carrying the homoplasmic variant m.14674T>C. They concluded that the likelihood of co-occurrence of such a rare nuclear variant with homoplasmy for m.14674T>C was more consistent with the observed disease prevalence in their studied cohort and established functional links (see Figure 3b) [52].

To our knowledge, this is the first report to provide evidence for digenic interactions in a mitochondrial disorder in a larger cohort.

Another example is Leber hereditary optic neuropathy (LHON), the most frequent mitochondrial disorder, characterized by gender-specific reduced penetrance. Recent studies estimate the penetrance of LHON to be significantly lower than the frequently cited 10% in females and 50% in males, at about 1.11–2.5% overall [53,54,55].

Molecularly, LHON is caused by mtDNA or nuclear variants impacting complex I function. Smoking and excessive alcohol consumption have been identified as lifestyle risk factors to manifest with LHON [56]. However, lifestyle factors alone are unable to fully explain the phenomenon of reduced penetrance in LHON. Further features, such as presently undetermined modifying genetic factors, could play a role.

### 4.3. Digenic Inheritance in LHON/Leigh Spectrum Disorder

In patients with classical LHON mutations, carriers of the most common mtDNA variant, m.11778G>A, very rare cases of manifestations with a Leigh spectrum phenotype have been reported [57,58,59].

Initial evidence for a digenic background of the LHON/Leigh spectrum was reported by Stenton et al. [60]. Stenton et al. identified pathogenic variants in the nuclear *DNAJC30* gene as the cause of autosomal recessive LHON. They were able to show that DNAJC30 is involved in complex I repair. In their cohort of patients with biallelic pathogenic *DNAJC30* variants, patients who did not exhibit LHON but had a Leigh spectrum disorder stood out. A systematic analysis of genes coding for complex I subunits led to the finding of heterozygous, rare high-impact *NDUFS2* and *NDUFS8* variants co-occurring with biallelic pathogenic DNAJC30 variants in patients with Leigh phenotypes (see Figure 3c) [61]. This concept was further confirmed by similar findings of heterozygous *NDUFS8* or *NDUFA9* variants with *DNAJC30*-associated Leigh cases [62,63]. Recently, more digenic Leigh phenotypes were reported in patients with the classic mitochondrial LHON variant m.11778A>G and heterozygous variants in *NDUFS2*, *NDUFS7*, and *NDUFS8* [64].

Further cases with possible penetrance-modifying nuclear and mitochondrial digenic mechanisms have been reported, such as a *VARS2*/MT-*TV* (m.1630A>G) in a proband with a MELAS phenotype [65] and *YARS*/MT-*ND1*(m.3635G>A) in patients with LHON [66]. 

### 4.4. Approaches and Challenges Investigating Digenic Inheritance in Mitochondriopathies

With WES and WGS as the diagnostic gold standard for mitochondrial disorders and a growing body of knowledge, the frequency of possible digenic findings will increase and further expand our pathomechanistic understanding of mitochondrial disorders. However, critical evaluation, including statistical analyses and an understanding of the challenges when addressing disease complexity in mitochondriopathies, will be paramount to put findings into perspective. 

The suspicion of a digenic interaction may arise from an observation in a single case or few pedigrees. A strong argument supporting such a finding is the replication of the observation in an independent cohort. Mitochondriopathies are highly heterogeneous and very rare diseases, diminishing the likelihood to observe the occurrence of a gene–gene pair in a digenic case multiple times for validation in a small cohort. This is best overcome by combining efforts through cooperation, for example, through the building of a global registry such as GENOMIT, and by sharing information on observations in databases like OLIDA, to document possible interactions and the level of evidence.

The examples we know of digenic inheritance in mitochondrial diseases have the following in common: by rare disease standards, they each involve a relatively frequent variant (m.14674T>C: MAF ~ 0.005; *DNAJC30* p.Tyr51Cys: MAF ~ 0.002; m.11778G>A: MAF ~ 0.0002), usually homoplasmic in the mtDNA, that interacts with a rare nuclear coded variant in a component from the same pathway or protein complex. Both properties reduce the search space and thereby increase the power to find statistical evidence. Another factor is phenotype variability for disease modifying factors. The more homogeneous the clinical phenotype, the higher the chance to discover clusters of distinct phenotypes associated with the same variant or variants in the same gene which could be indicative of a second genetic factor, such as in Leigh syndrome, in patients with LHON-causing variants. It is a great challenge to find evidence of digenic interactions between ultra-rare variants and between genes where a functional link has not yet been established. 

One of these hard to interpret interactors are epigenetic modifiers. With the essential role of mitochondrial function in energy metabolism, it is unsurprising that they are subject to regulation for an adequate answer to various external and metabolic circumstances. However, the mediating mechanisms of these effects and their role, especially in the mitochondrial genome, are not fully understood and subject to ongoing research. Whether, or to what extent, mtDNA is methylated remains subject to controversial debate [67]. Still, an isoform of DNA methyltransferase 1 (DNMT1) appears to be able to translocate into mitochondria and bind to mtDNA, providing a plausible mechanism for cytosine methylation in mtDNA and implicating epigenetic mechanisms as regulatory processes also in mitochondrial function and disorders [68,69,70]. Epigenetic modifiers and their contribution need to be further characterized to be better understood in a complex disease model. This is especially challenging considering the interplay of mitochondrial and nuclear genome. A well-researched example of such interaction is mitochondrial tRNA modification by nuclear coded enzymes, such as *TRMU* or *EARS2*, which have been linked to a digenic disease mechanism in RIRDC with the mitochondrial variant m.14674T>C [52]. Epigenetic modification of mitochondrial RNA is a well-established mechanism to optimize protein translation [71]. Another example of a disease-relevant well-described modifier is *NSUN3*, a methyltransferase that modifies a specific cytosine (C34) of the mitochondrial tRNA for methionine. Biallelic pathogenic variants in *NSUN3* are causative of combined oxidative phosphorylation deficiency, further underlining the biological relevance of these nuclear epigenetic modifiers for mitochondrial function [72]. 

## 5. Outlook

In summary, the first efforts to detect digenic interactions in mitochondrial diseases have already been successful and support that digenicity plays a role in mitochondriopathies. The first findings so far are mostly associated with digenic modifiers explaining phenotypic variability and reduced penetrance in well-characterized mitochondrial diseases. We propose to approach the question of digenic interactions in mitochondriopathies through the lens of functional interactions within unsolved, clinically convincing cases. 

Based on the existing experience, investigating digenic modifiers in mitochondrial disorders holds most promise in diseases with remarkably reduced penetrance, such as LHON and RIRDC, or in disorders showing strong phenotypic variability, such as MELAS. Approaches focusing on mtDNA variants’ digenic interaction with a known nuclear gene appear auspicious and can build on pre-existing knowledge of functional interaction. Modifying enzymes and proteins to optimize or regulate molecular processes in mitochondria are of particular interest, as exemplified by *DNAJC30*, a chaperone protein that facilitates Complex I repair, and *TRMU*, a mitochondrial tRNA modifier.

Furthermore, as a basis to address the question of complex inheritance in mitochondrial disorders, it is necessary to build on large population-based studies and disease specific cohorts, which are best achieved through international collaboration and large consortia such as GENOMIT. Novel statistical methods and computational prediction tools are necessary to verify the findings leveraging large datasets.

The digenic disease model likely presents, to some extent, another simplification in genetics. In contrast to frequent disorders, where multifactorial disease backgrounds can be explored by GWAS analysis, rare disorders necessitate different strategies to identify genetic patterns beyond the monogenic disease model. 

So far, the detection of digenic inheritance is limited to cases with the involvement of at least one gene known to cause a monogenic rare disease. As these genetic variants are, per definition, rare, a digenic combination is usually extremely rare and therefore hard to prove statistically in the absence of sufficiently large cohorts. As a combination of defects in genes, that are not linked to a monogenic disease, could also interact to cause a digenic disorder, novel approaches are necessary. Individual contributing genetic variants may then show a higher allele frequency, and statistical approaches may be more suitable for validation thereafter.

When looking at digenic interaction on a functional level, to date, mostly additive detrimental factors are being considered. It is, however, likely that protective genetic factors are also relevant to shape a phenotype. These may be of particular interest with regard to therapy development but remains challenging to investigate, as due to their beneficial properties, they are not subject to constraint and therefore not expected to be rare.

We surmise that investigation of digenic interplay can be a first step towards a better understanding of the complexities in hereditary disorders in general and in mitochondriopathies in specific. A thorough understanding of disease mechanisms and of exacerbating or beneficial contributing genetic effects can in turn impact on the diagnostic yield, prognostic strength and, ultimately, therapy and management of a disorder. 

## Figures and Tables

**Figure 1 ijms-25-04602-f001:**
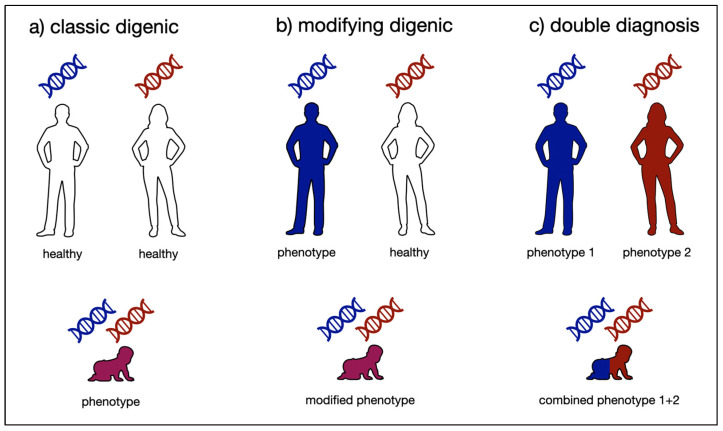
Schematic overview of inheritance mechanisms involving the contribution of damaging variants in two genes. (**a**) Classic digenic inheritance: both healthy parents carry a genetic aberration in a distinct gene; if offspring inherits both genetic changes, a phenotype develops due to functional interaction of both genetic changes; (**b**) modifying digenic inheritance: one parent carries a genetic aberration causing a phenotype, while the other parent carries a genetic aberration in another gene without showing a phenotype; an offspring inheriting both changes shows a phenotype distinct from the affected parent, modified by the genetic change inherited from the other healthy parent; (**c**) double diagnosis: both parents show distinct phenotypes due to distinct genetic changes, and offspring inheriting both changes shows a combined phenotype with symptoms from both disorders.

**Figure 2 ijms-25-04602-f002:**
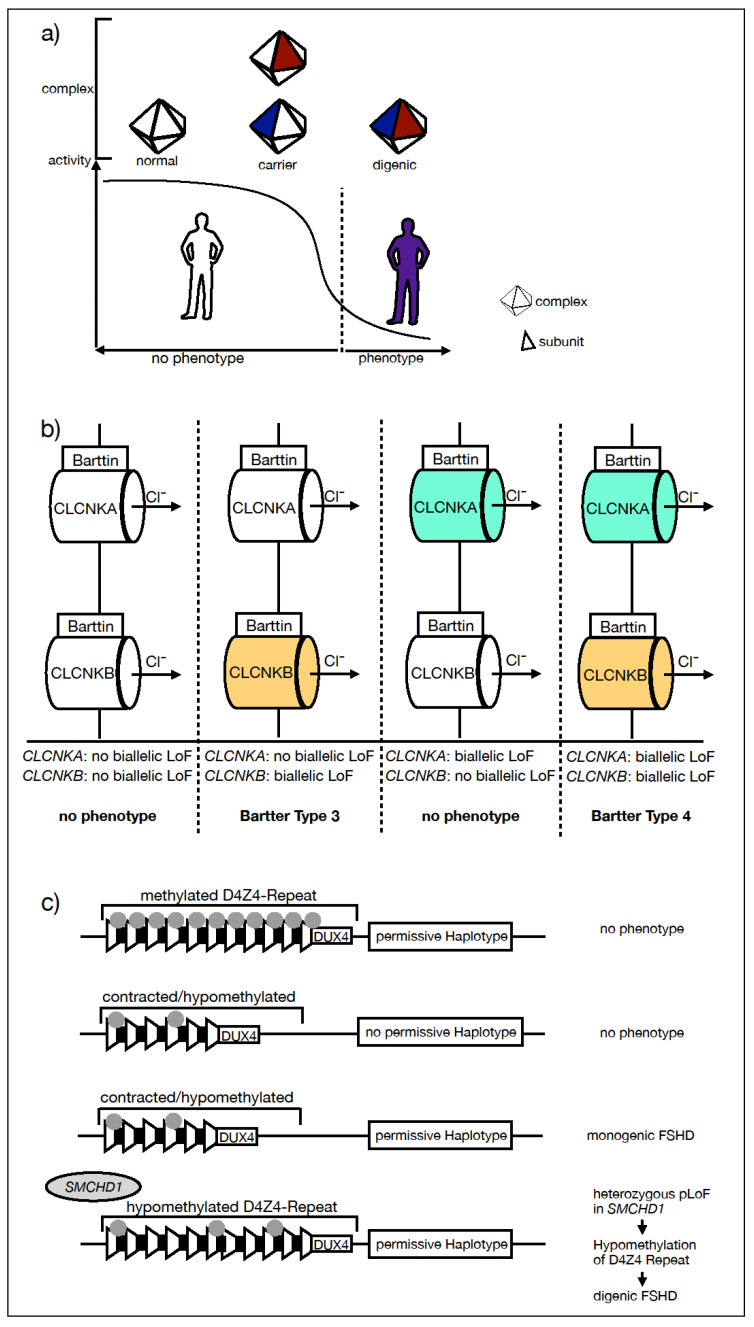
Pathomechanisms of known digenic disorders. (**a**) Schematic representation of protein–protein interaction leading to a protein complex dysfunction; (**b**) indirect gene product interaction exemplified by Bartter syndrome type 4 as a result of biallelic LoF variants in *CLCNKB* and *CLCNKA*; (**c**) FSHD as an example of epigenetic modification in digenic disease.

**Figure 3 ijms-25-04602-f003:**
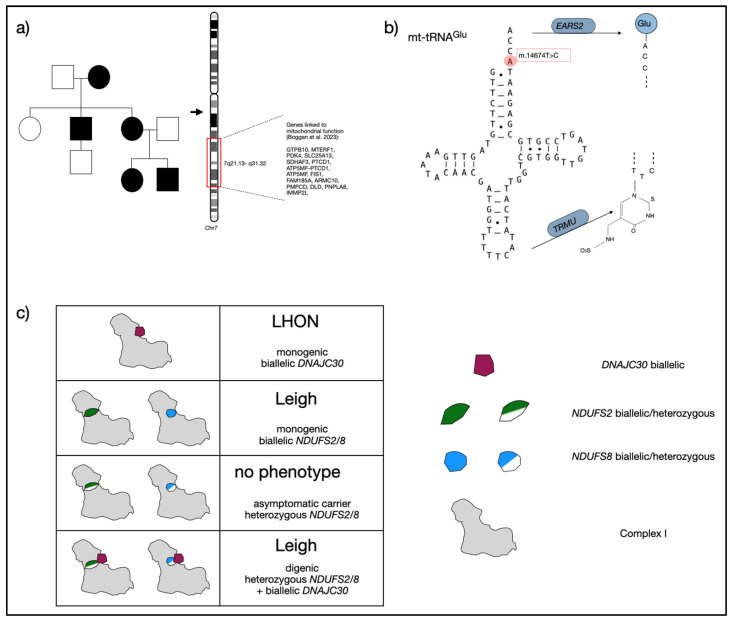
Digenic inheritance and disease mechanisms reported in mitochondrial disorders. (**a**) Schematic representation of an approach using linkage analysis in larger families to detect genetic modifying factors impacting a phenotype. Here, we chose the example of encephalopathy as a phenotypic trait in m.3243A>G associated MELAS syndrome, which was found to be genetically linked to chromosomal region 7q22 by Boggan et al. [50]. (**b**) Visualization of functional link between mt-tRNA^Glu^ and tRNA modifiers *EARS2*, which codes for the glutamyl-tRNA-synthetase 2 and *TRMU*, coding for the tRNA 5-methylaminomethyl-2-thiouridylate methyltransferase and thus modifying all mitochondrial tRNAs for optimal translation. (**c**) Digenic Leigh syndrome in patients with biallelic *DNAJC30* and heterozygous *NDUFS2*/*8* variants; while biallelic *NDUFS2*/*8* variants are associated with Leigh syndrome, biallelic *DNAJC30* variants cause LHON, and heterozygous *NDUFS2*/*8* or *DNAJC30* variants alone do not cause a phenotype. DNAJC30 is a complex I assembly factor, while *NDUFS2*/*8* are subunits of complex I.

## Data Availability

No new data were created or analyzed in this study. Data sharing is not applicable to this article.

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
