# Peer review of "Digenic Inheritance in Rare Disorders and Mitochondrial Disease—Crossing the Frontier to a More Comprehensive Understanding of Etiology"

_ijms, 2024, doi:10.3390/ijms25094602_

Round 1

Reviewer 1 Report

Comments and Suggestions for Authors

This review discusses the details of digenic inheritance, its pathomechanisms, and approaches and challenges involved in investigating digenic inheritance. Additionally, authors have also reviewed the monogenic inheritance in a similar way. Further, authors have reviewed the details and prospects of digenic inheritance in mitochondrial disorders.

Comments:

1.       Please include a brief section of non-coding variants/mutations involved in monogenic disorders to make it more comprehensive.

2.       Please cite the appropriate references in the introduction section of the manuscript.

3.       Please remove the web link (https://www.ebi.ac.uk/GWAS/) from the table 1 since there are other GWAS data available from different populations. ‘GWAS Catalog’ should be sufficient in this section.

4.       Please provide sub-sections for section 4 (Digenic inheritance in mitochondrial disorders) such as “Pathomechanisms of digenic inheritance in mitochondrial disorders”, “Investigating digenic inheritance in mitochondrial disorders: approaches and challenges” etc.

5.       As there is no space problem for this pedigree in figure 3a, please use the conventional lines for pedigree in figure 3a as illustrated below instead of downward lines for I-1 & I-2 and II-3 & II-4.

Best regards

Author Response

Review 1:

We would like to thank you very much for the time and effort invested in reviewing our manuscript and for your suggestions. We will be addressing your comments one by one:

This review discusses the details of digenic inheritance, its pathomechanisms, and approaches and challenges involved in investigating digenic inheritance. Additionally, authors have also reviewed the monogenic inheritance in a similar way. Further, authors have reviewed the details and prospects of digenic inheritance in mitochondrial disorders.

Comments:

  1. Please include a brief section of non-coding variants/mutations involved in monogenic disorders to make it more comprehensive.

Response: Thank you for your comment and suggestion. While we fully agree on the importance of the topic of non-coding variants in monogenic disorders, we have decided against including a more thorough exploration of their role in monogenic disease within the scope of this review. We aimed at including variant interpretation under a monogenic model as basis to understand variant assessment in a digenic setting, but did not have the space that would have been needed to provide a fully comprehensive compendium of variant interpretation.  Therefore, we did not aim at completeness but at providing a monogenic perspective before thinking about variant assessment to address the question of digenic inheritance. Topics such as the functional consequence of variants, non-coding variants and CNVs were therefore omitted.

  1. Please cite the appropriate references in the introduction section of the manuscript.

Response: Thank you for your suggestion. We have added the following references to further support statements made in the introduction:

Bamshad, Michael J, Sarah B Ng, Abigail W Bigham, Holly K Tabor, Mary J Emond, Deborah A Nickerson, and Jay Shendure. "Exome Sequencing as a Tool for Mendelian Disease Gene Discovery." Nature Reviews Genetics 12, no. 11 (2011): 745-55.

Chakravarti, Aravinda. "Magnitude of Mendelian Versus Complex Inheritance of Rare Disorders." American Journal of Medical Genetics Part A 185, no. 11 (2021): 3287-93.

Clark, Michelle M, Zornitza Stark, Lauge Farnaes, Tiong Y Tan, Susan M White, David Dimmock, and Stephen F Kingsmore. "Meta-Analysis of the Diagnostic and Clinical Utility of Genome and Exome Sequencing and Chromosomal Microarray in Children with Suspected Genetic Diseases." NPJ genomic medicine 3, no. 1 (2018): 16.

Yang, Yaping, Donna M Muzny, Jeffrey G Reid, Matthew N Bainbridge, Alecia Willis, Patricia A Ward, Alicia Braxton, Joke Beuten, Fan Xia, and Zhiyv Niu. "Clinical Whole-Exome Sequencing for the Diagnosis of Mendelian Disorders." New England Journal of Medicine 369, no. 16 (2013): 1502-11.

Wright, Caroline F, Jeremy F McRae, Stephen Clayton, Giuseppe Gallone, Stuart Aitken, Tomas W FitzGerald, Philip Jones, Elena Prigmore, Diana Rajan, and Jenny Lord. "Making New Genetic Diagnoses with Old Data: Iterative Reanalysis and Reporting from Genome-Wide Data in 1,133 Families with Developmental Disorders." Genetics in Medicine 20, no. 10 (2018): 1216-23.

Gazzo, A., D. Raimondi, D. Daneels, Y. Moreau, G. Smits, S. Van Dooren, and T. Lenaerts. "Understanding Mutational Effects in Digenic Diseases." Nucleic Acids Res 45, no. 15 (2017): e140.

Posey, J. E., T. Harel, P. Liu, J. A. Rosenfeld, R. A. James, Z. H. Coban Akdemir, M. Walkiewicz, W. Bi, R. Xiao, Y. Ding, F. Xia, A. L. Beaudet, D. M. Muzny, R. A. Gibbs, E. Boerwinkle, C. M. Eng, V. R. Sutton, C. A. Shaw, S. E. Plon, Y. Yang, and J. R. Lupski. "Resolution of Disease Phenotypes Resulting from Multilocus Genomic Variation." N Engl J Med 376, no. 1 (2017): 21-31.

  1. Please remove the web link (https://www.ebi.ac.uk/GWAS/) from the table 1 since there are other GWAS data available from different populations. ‘GWAS Catalog’ should be sufficient in this section.

Response: We have removed the link from the table.

  1. Please provide sub-sections for section 4 (Digenic inheritance in mitochondrial disorders) such as “Pathomechanisms of digenic inheritance in mitochondrial disorders”, “Investigating digenic inheritance in mitochondrial disorders: approaches and challenges” etc.

Response: Thank you for pointing this out. We have added subsections to improve the structure of this section.

  1. As there is no space problem for this pedigree in figure 3a, please use the conventional lines for pedigree in figure 3a as illustrated below instead of downward lines for I-1 & I-2 and II-3 & II-4.

Response: Thank you for pointing this out. We have changed the pedigree in figure 3a accordingly.

Best regards

Thank you for reviewing our manuscript. We look forward to hearing your feedback regarding our resubmission and will be happy to respond to any further questions and comments you may have.

Reviewer 2 Report

Comments and Suggestions for Authors

I hope this letter finds you well. I would like to express my gratitude for having an opportunity to review a manuscript entitled Digenic inheritance in mitochondrial disease – crossing the frontier to a more comprehensive understanding of etiology to our journal. Although I found myself rather excited and enthusiastic throughout the review process, I believe, and thus regret to inform you, that the manuscript needs several amendments before it can be accepted for publication. However, I believe that with some minor revisions, your manuscript has the potential to make a significant contribution to the field. Thank you very much for the opportunity and your work.

Overall strengths of the review:

·         Thorough Exploration: The manuscript provides a comprehensive exploration of digenic inheritance in rare disorders, particularly focusing on mitochondrial diseases.

·         Clear Presentation: Concepts related to digenic inheritance are well-explained, making the manuscript accessible to readers with varying levels of expertise.

·         Examples and Case Studies: The inclusion of examples of digenic inheritance in mitochondrial disorders adds depth to the discussion and enhances understanding.

·         Detailed Examination of Approaches: The manuscript thoroughly examines approaches and challenges in investigating digenic inheritance, providing valuable insights into variant evaluation, statistical analysis, and functional characterization.

Overall weaknesses of the review:

·         Lack of Conciseness: Some sections of the manuscript are overly detailed, which could potentially overwhelm readers.

·         Organization: The flow of the manuscript could be improved for better coherence and readability.

·         Clarity in Figures: The figures could be clearer (e.g., more informative) and better integrated with the text to enhance understanding.

·         Limited Discussion on Epigenetic Modifiers: The discussion on epigenetic modifiers in mitochondrial disorders is brief and could be expanded to provide a more comprehensive overview.

Recommendations for Improvements:

1.      Streamline Content: Condense overly detailed sections to maintain reader engagement and improve overall readability.

2.      Enhance Organization: Re-evaluate the structure of the manuscript to ensure a logical flow of ideas and concepts.

3.      Revise Figures: Clarify and improve the quality of figures to aid in understanding and better integration with the text.

4.      Expand Discussion on Epigenetic Modifiers: Provide a more in-depth discussion on the role of epigenetic modifiers in mitochondrial disorders to offer a comprehensive perspective.

5.      Include Future Directions: Offer specific recommendations for future research directions and potential areas of exploration in the field of digenic inheritance.

Comments on the Quality of English Language

Proofreading: Conduct thorough proofreading to address any grammatical errors or inconsistencies in language usage (e.g. mix of American and English spelling, passive and active verbs, inconsistent usage of commas and punctuation, local awkward phrasing etc.).

Author Response

Review 2:

We would like to thank you very much for reviewing our manuscript and for your suggestions. We will be addressing your comments one by one:

I hope this letter finds you well. I would like to express my gratitude for having an opportunity to review a manuscript entitled Digenic inheritance in mitochondrial disease – crossing the frontier to a more comprehensive understanding of etiology to our journal. Although I found myself rather excited and enthusiastic throughout the review process, I believe, and thus regret to inform you, that the manuscript needs several amendments before it can be accepted for publication. However, I believe that with some minor revisions, your manuscript has the potential to make a significant contribution to the field. Thank you very much for the opportunity and your work.

Overall strengths of the review:

  • Thorough Exploration: The manuscript provides a comprehensive exploration of digenic inheritance in rare disorders, particularly focusing on mitochondrial diseases.
  • Clear Presentation: Concepts related to digenic inheritance are well-explained, making the manuscript accessible to readers with varying levels of expertise.
  • Examples and Case Studies: The inclusion of examples of digenic inheritance in mitochondrial disorders adds depth to the discussion and enhances understanding.
  • Detailed Examination of Approaches: The manuscript thoroughly examines approaches and challenges in investigating digenic inheritance, providing valuable insights into variant evaluation, statistical analysis, and functional characterization.

Thank you very much for pointing out these strengths of our manuscript!

Overall weaknesses of the review:

  • Lack of Conciseness: Some sections of the manuscript are overly detailed, which could potentially overwhelm readers.
  • Organization: The flow of the manuscript could be improved for better coherence and readability.
  • Clarity in Figures: The figures could be clearer (e.g., more informative) and better integrated with the text to enhance understanding.
  • Limited Discussion on Epigenetic Modifiers: The discussion on epigenetic modifiers in mitochondrial disorders is brief and could be expanded to provide a more comprehensive overview.

Recommendations for Improvements:

  1. Streamline Content: Condense overly detailed sections to maintain reader engagement and improve overall readability.

Response: We have condensed the text especially in section 3.3. Together with improved structure (point 2) we are convinced that this improves the overall readability.

  1. Enhance Organization: Re-evaluate the structure of the manuscript to ensure a logical flow of ideas and concepts.

Response: Thank you for this observation. To improve the structure, we added sub-sections in chapter 2 and 4 and revised chapter 4, adding further detail on digenic inheritance in LHON and MELAS.

  1. Revise Figures: Clarify and improve the quality of figures to aid in understanding and better integration with the text.

Response: To improve integration of the figures in the text and for clarity descriptive captions have been added.

  1. Expand Discussion on Epigenetic Modifiers: Provide a more in-depth discussion on the role of epigenetic modifiers in mitochondrial disorders to offer a comprehensive perspective.

Response: Thank you for your suggestion. We have elaborated the discussion on epigenetic modifiers and their role in mitochondrial disorders further and hope to provide a more comprehensive view of the topic now.

  1. Include Future Directions: Offer specific recommendations for future research directions and potential areas of exploration in the field of digenic inheritance.

Response: Thank you for raising this point. We added a paragraph at the end of the chapter on digenic inheritance in mitochondrial disorders deriving specific recommendations for research into this area from the experiences in the field so far.  

Comments on the Quality of English Language

Proofreading: Conduct thorough proofreading to address any grammatical errors or inconsistencies in language usage (e.g. mix of American and English spelling, passive and active verbs, inconsistent usage of commas and punctuation, local awkward phrasing etc.).

We have made an effort to improve the English language.

Thank you for the time and effort invested in reviewing our manuscript. We look forward to hearing your feedback regarding our resubmission and will be happy to respond to any further questions and comments you may have.

Reviewer 3 Report

Comments and Suggestions for Authors

As an external reviewer, I have carefully examined your manuscript titled "Digenic inheritance in mitochondrial disease – crossing the frontier to a more comprehensive understanding of etiology." Below are my comments and suggestions for improvement:

  1. While the review offers informative insights into the definition of digenic, monogenic, and multifactorial diseases, the section on mitochondrial disease seems somewhat superficial and could benefit from further description and development.

  2. - The title appears overly specific to mitochondrial disease, yet the review does not entirely focus on mitochondria. I suggest either adjusting the title to offer a more general description of the paper's content or incorporating more information about known mitochondrial diseases and digenic inheritance as a potential pathogenic mechanism.

  3. - Please define the acronyms WES or WGS when first mentioned (Line 37).

  4. - It's advisable to provide citations for statements made in Lines 47-50.

  5. - Regarding Figure 1, while the figure is straightforward, it may be challenging to follow upon initial inspection. I recommend adding information indicating that healthy parents carry a gene variant, or alternatively, enhancing the explanation in the figure caption.

  6. - Define the acronym LoF (Line 92).

  7. - Similarly, define the acronym pLoF (Line 98).

  8. - There appears to be a typographical error in Line 112; "DZ4Z" should be corrected to "D4Z4."

  9. - Please correct the phrase "Both these factors" in Line 187 for clarity.

  10. - Define the acronym MAFs in Line 209.

  11. - There is a typographical error in Line 303; "cooccuring" should be corrected to "co-occurring."

  12. - In Line 394, "homegenous" should be corrected to "homogeneous."

- I believe addressing these points will significantly improve the quality and clarity of your manuscript. Thank you for considering my feedback, and I look forward to seeing the revised version.

Comments on the Quality of English Language

Overall, I find the English language usage satisfactory. However, I would like to highlight the need for correcting typographical errors and providing definitions for acronyms throughout the manuscript.

Author Response

Review Nr. 3

We would like to thank you for reviewing our manuscript, for your suggestions and the opportunity to resubmit a revised draft of our manuscript. We will be addressing your comments one by one:

As an external reviewer, I have carefully examined your manuscript titled "Digenic inheritance in mitochondrial disease – crossing the frontier to a more comprehensive understanding of etiology." Below are my comments and suggestions for improvement:

While the review offers informative insights into the definition of digenic, monogenic, and multifactorial diseases, the section on mitochondrial disease seems somewhat superficial and could benefit from further description and development.

- The title appears overly specific to mitochondrial disease, yet the review does not entirely focus on mitochondria. I suggest either adjusting the title to offer a more general description of the paper's content or incorporating more information about known mitochondrial diseases and digenic inheritance as a potential pathogenic mechanism.

Response: We agree the title may be misleading, raising the expectation of a stronger focus on mitochondrial disorders. We aimed for a thorough overview over the topic of digenic inheritance in rare hereditary disorders and explore if these concepts apply in mitochondrial disorders using examples of known and suspected digenic mechanisms in the field. To hopefully better reflect the more general part of the review we changed the title to: Digenic inheritance in rare disorders and mitochondrial disease – crossing the frontier to a more comprehensive understanding of etiology

- Please define the acronyms WES or WGS when first mentioned (Line 37).

Response: Thank you for pointing this out, we defined the acronyms as suggested.

- It's advisable to provide citations for statements made in Lines 47-50.

Response: Citations have been added to support the definition of digenic inheritance and modifying inheritance and to support the distinction from double diagnosis: Gazzo, Andrea, et al. "Understanding mutational effects in digenic diseases." Nucleic acids research 45.15 (2017): e140-e140. and Posey, Jennifer E., et al. "Resolution of disease phenotypes resulting from multilocus genomic variation." New England Journal of Medicine 376.1 (2017): 21-31.

- Regarding Figure 1, while the figure is straightforward, it may be challenging to follow upon initial inspection. I recommend adding information indicating that healthy parents carry a gene variant, or alternatively, enhancing the explanation in the figure caption.

Response: We have added explanatory text to make this figure and its intended message easier to understand.

- Define the acronym LoF (Line 92).

- Similarly, define the acronym pLoF (Line 98).

- There appears to be a typographical error in Line 112; "DZ4Z" should be corrected to "D4Z4."

- Please correct the phrase "Both these factors" in Line 187 for clarity.

- Define the acronym MAFs in Line 209.

- There is a typographical error in Line 303; "cooccuring" should be corrected to "co-occurring."

- In Line 394, "homegenous" should be corrected to "homogeneous."

Response: Thank you for pointing these issues out - we defined the acronyms and corrected the errors as suggested. The sentence in line 187 has been adapted for more clarity.

- I believe addressing these points will significantly improve the quality and clarity of your manuscript. Thank you for considering my feedback, and I look forward to seeing the revised version.

Comments on the Quality of English Language

Overall, I find the English language usage satisfactory. However, I would like to highlight the need for correcting typographical errors and providing definitions for acronyms throughout the manuscript.

Thank you for the time and effort invested in reviewing our manuscript. We look forward to hearing your feedback regarding our resubmission and will be happy to respond to any further questions and comments you may have.